# MoËT: Interpretable & Verifiable Reinforcement Learning via Mixture of Expert Trees

## Abstract

Deep Reinforcement Learning (DRL) has led to many recent breakthroughs on complex control tasks, such as defeating the best human player in the game of Go. However, decisions made by the DRL agent are not explainable, hindering its applicability in safety-critical settings. Viper, a recently proposed technique, constructs a decision tree policy by mimicking the DRL agent. Decision trees are interpretable as each action made can be traced back to the decision rule path that lead to it. However, one global decision tree approximating the DRL policy has significant limitations with respect to the geometry of decision boundaries. We propose MoËT, a more expressive, yet still interpretable model based on Mixture of Experts, consisting of a gating function that partitions the state space, and multiple decision tree experts that specialize on different partitions. We propose a training procedure to support non-differentiable decision tree experts and integrate it into imitation learning procedure of Viper. We evaluate our algorithm on four OpenAI gym environments, and show that the policy constructed in such a way is more performant and better mimics the DRL agent by lowering mispredictions and increasing the reward. We also show that MoËT policies are amenable for verification using off-the-shelf automated theorem provers such as Z3.

## 1 Introduction

Deep Reinforcement Learning (DRL) has achieved many recent breakthroughs in challenging domains such as Go (Silver et al., 2016). While using neural networks for encoding state representations allow DRL agents to learn policies for tasks with large state spaces, the learned policies are not interpretable, which hinders their use in safety-critical applications.

Some recent works leverage programs and decision trees as representations for interpreting the learned agent policies. Pirl(Verma et al., 2018) uses program synthesis to generate a program in a Domain-Specific Language (DSL) that is close to the DRL agent policy. The design of the DSL with desired operators is a tedious manual effort and the enumerative search for synthesis is difficult to scale for larger programs. In contrast, Viper (Bastani et al., 2018) learns a Decision Tree (DT) policy by mimicking the DRL agent, which not only allows for a general representation for different policies, but also allows for verification of these policies using integer linear programming solvers.

Viper uses the Dagger (Ross et al., 2011) imitation learning approach to collect state action pairs for training the student DT policy given the teacher DRL policy. It modifies the Dagger algorithm to use the Q-function of teacher policy to prioritize states of critical importance during learning. However, learning a single DT for the complete policy leads to some key shortcomings such as i) less faithful representation of original agent policy measured by the number of mispredictions, ii) lower overall performance (reward), and iii) larger DT sizes that make them harder to interpret.

In this paper, we present MoËT (Mixture of Expert Trees), a technique based on Mixture of Experts (MoE) (Jacobs et al., 1991; Jordan and Xu, 1995; Yuksel et al., 2012), and reformulate its learning procedure to support DT experts. MoE models can typically use any expert as long as it is a differentiable function of model parameters, which unfortunately does not hold for DTs. Similar to MoE training with Expectation-Maximization (EM) algorithm, we first observe that MoËT can be trained by interchangeably optimizing the weighted log likelihood for experts (independently from one another) and optimizing the gating function with respect to the obtained experts. Then, we propose a procedure for DT learning in the specific context of MOE. To the best of our knowledge we are first to combine standard non-differentiable DT experts, which are interpretable, with MoE

model. Existing combinations which rely on differentiable tree or treelike models, such as soft decision trees (Irsoy et al., 2012) and hierarchical mixture of experts (Zhao et al., 2019) are not interpretable.

We adapt the imitation learning technique of Viper to use MOËT policies instead of DTs. MOËT creates multiple local DTs that specialize on different regions of the input space, allowing for simpler (shallower) DTs that more accurately mimic the DRL agent policy within their regions, and combines the local trees into a global policy using a gating function. We use a simple and interpretable linear model with softmax function as the gating function, which returns a distribution over DT experts for each point in the input space. While standard MOE uses this distribution to average predictions of DTs, we also consider selecting just one most likely expert tree to improve interpretability. While decision boundaries of Viper DT policies must be axis-perpendicular, the softmax gating function supports boundaries with hyperplanes of arbitrary orientations, allowing MOËT to more faithfully represent the original policy.

We evaluate our technique on four different environments: CartPole, Pong, Acrobot, and Mountaincar. We show that MOËT achieves significantly better rewards and lower misprediction rates with shallower trees. We also visualize the Viper and MOËT policies for Mountaincar, demonstrating the differences in their learning capabilities. Finally, we demonstrate how a MOËT policy can be translated into an SMT formula for verifying properties for CartPole game using the Z3 theorem prover (De Moura and Bjørner, 2008) under similar assumptions made in Viper.

In summary, this paper makes the following key contributions: 1) We propose MOËT, a technique based on MOE to learn mixture of expert decision trees and present a learning algorithm to train MOËT models. 2) We use MOËT models with a softmax gating function for interpreting DRL policies and adapt the imitation learning approach used in Viper to learn MOËT models. 3) We evaluate MOËT on different environments and show that it leads to smaller, more faithful, and performant representations of DRL agent policies compared to Viper while preserving verifiability.

## 2 RELATED WORK

*Interpretable Machine Learning:* In numerous contexts, it is important to understand and interpret the decision making process of a machine learning model. However, interpretability does not have a unique definition that is widely accepted. Accoding to Lipton (Lipton, 2016), there are several properties which might be meant by this word and we adopt the one which Lipton names *transparency* which is further decomposed to *simulability*, *decomposability*, and *algorithmic transparency*. A model is simulable if a person can in reasonable time compute the outputs from given inputs and in that way simulate the model's inner workings. That holds for small linear models and small decision trees (Lipton, 2016). A model is decomposable if each part of a models admits an intuitive explanation, which is again the case for simple linear models and decision trees (Lipton, 2016). Algorithmic transparency is related to our understanding of the workings of the training algorithm. For instance, in case of linear models the shape of the error surface and properties of its unique minimum towards which the algorithm converges are well understood (Lipton, 2016). MOËT models focus on transparency (as we discuss at the end of Section 5).

*Explainable Machine Learning:* There has been a lot of recent interest in explaining decisions of black-box models (Guidotti et al., 2018a; Doshi-Velez and Kim, 2017). For image classification, activation maximization techniques can be used to sample representative input patterns (Erhan et al., 2009; Olah et al., 2017). TCAV (Kim et al., 2017) uses human-friendly high-level concepts to associate their importance to the decision. Some recent works also generate contrastive robust explanations to help users understand a classifier decision based on a family of neighboring inputs (Zhang et al., 2018; Dhurandhar et al., 2018). LORE (Guidotti et al., 2018b) explains behavior of a black-box model around an input of interest by sampling the black-box model around the neighborhood of the input, and training a local DT over the sampled points. Our model presents an approach that combines local trees into a global policy.

*Tree-Structured Models:* Irsoy et al. (Irsoy et al., 2012) propose a a novel decision tree architecture with soft decisions at the internal nodes where both children are chosen with probabilities given by a sigmoid gating function. Similarly, binary tree-structured hierarchical routing mixture of experts (HRME) model, which has classifiers as non-leaf node experts and simple regression models as leaf node experts, were proposed in (Zhao et al., 2019). Both models are unfortunately not interpretable.

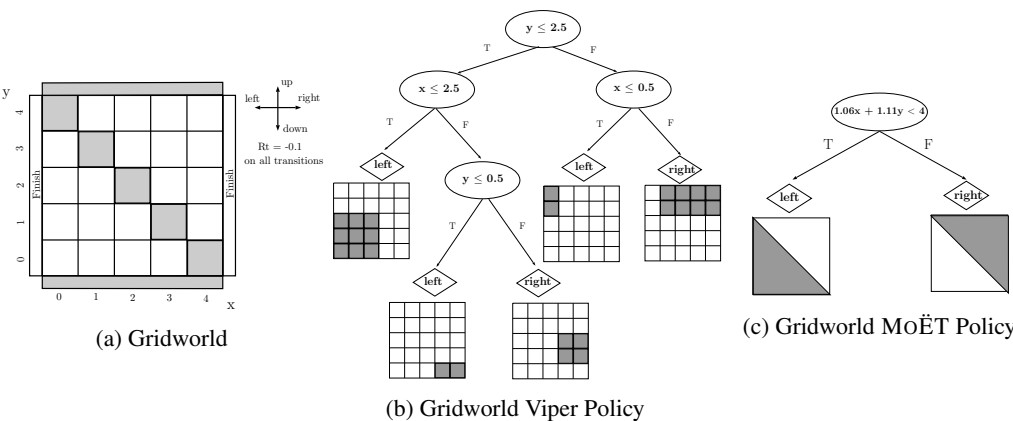

Figure 1: A simple 5 x 5 Gridworld example to showcase the policies learned by Viper and MoËT.

*Knowledge Distillation and Model Compression:* We rely on ideas already explored in fields of model compression (Bucilu et al., 2006) and knowledge distillation (Hinton et al., 2015). The idea is to use a complex well performing model to facilitate training of a simpler model which might have some other desirable properties (e.g., interpretability). Such practices have been applied to approximate decision tree ensemble by a single tree (Breiman and Shang, 1996), but this is different from our case, since we approximate a neural network. In a similar fashion a neural network can be used to train another neural network (Furlanello et al., 2018), but neural networks are hard to interpret and even harder to formally verify, so this is also different from our case. Such practices have also been applied in the field of reinforcement learning in knowledge and policy distillation (Rusu et al., 2016; Koul et al., 2019; Zhang et al., 2019), which are similar in spirit to our work, and imitation learning (Bastani et al., 2018; Ross et al., 2011; Abbeel and Ng, 2004; Schaal, 1999), which provide a foundation for our work.

## 3    MOTIVATING EXAMPLE: GRIDWORLD

We now present a simple motivating example to showcase some of the key differences between Viper and MoËT approaches. Consider the $N \times N$ Gridworld problem shown in Figure 1a (for $N = 5$). The agent is placed at a random position in a grid (except the walls denoted by filled rectangles) and should find its way out. To move through the grid the agent can choose to go up, left, right or down at each time step. If it hits the wall it stays in the same position (state). State is represented using two integer values ($x, y$ coordinates) which range from $(0, 0)$—bottom left to $(N - 1, N - 1)$—top right. The grid can be escaped through either left doors (left of the first column), or right doors (right of the last column). A negative reward of $-0.1$ is received for each agent action (negative reward encourages the agent to find the exit as fast as possible). An episode finishes as soon as an exit is reached or if 100 steps are made whichever comes first.

The optimal policy ($\pi_*$) for this problem consists of taking the left (right resp.) action for each state below (above resp.) the diagonal. We used $\pi_*$ as a teacher and used imitation learning approach of Viper to train an interpretable DT policy that mimics $\pi_*$. The resulting DT policy is shown in Figure 1b. The DT partitions the state space (grid) using lines perpendicular to x and y axes, until it separates all states above diagonal from those below. This results in a DT of depth 3 with 9 nodes. On the other hand, the policy learned by MoËT is shown in Figure 1c. The MoËT model with 2 experts learns to partition the space using the line defined by a linear function $1.06x + 1.11y = 4$ (roughly the diagonal of the grid). Points on the different sides of the line correspond to two different experts which are themselves DTs of depth 0 always choosing to go left (below) or right (above).

We notice that DT policy needs much larger depth to represent $\pi_*$ while MoËT can represent it as only one decision step. Furthermore, with increasing $N$ (size of the grid), complexity of DT will grow, while MoËT complexity stays the same; we empirically confirm this for $N = [5, 10]$. For $N = 5, 6, 7, 8, 9, 10$ DT depths are $3, 4, 4, 4, 4, 5$ and number of nodes are $9, 11, 13, 15, 17, 21$ respectively. In contrast, MoËT models of same complexity and structure as the one shown in Figure 1c are learned for all values of $N$ (models differ in the learned partitioning linear function).

## 4 BACKGROUND

In this section we provide description of two relevant methods we build upon: (1) Viper, an approach for interpretable imitation learning, and (2) MOE learning framework.

**Viper**. Viper algorithm (included in appendix) is an instance of DAGGER imitation learning approach, adapted to prioritize critical states based on Q-values. Inputs to the Viper training algorithm are (1) environment $e$ which is an finite horizon ($T$-step) Markov Decision Process (MDP) $(S, A, P, R)$ with states $S$, actions $A$, transition probabilities $P : S \times A \times S \rightarrow [0, 1]$, and rewards $R : S \rightarrow \mathbb{R}$; (2) teacher policy $\pi_t : S \rightarrow A$; (3) its Q-function $Q^{\pi_t} : S \times A \rightarrow \mathbb{R}$ and (4) number of training iterations $N$. Distribution of states after $T$ steps in environment $e$ using a policy $\pi$ is $d^{(\pi)}(e)$ (assuming randomly chosen initial state). Viper uses the teacher as an oracle to label the data (states with actions). It initially uses teacher policy to sample trajectories (states) to train a student (DT) policy. It then uses the student policy to generate more trajectories. Viper samples training points from the collected dataset $D$ giving priority to states $s$ having higher importance $I(s)$, where $I(s) = \max_{a \in A} Q^{\pi_t}(s, a) - \min_{a \in A} Q^{\pi_t}(s, a)$. This sampling of states leads to faster learning and shallower DTs. The process of sampling trajectories and training students is repeated for number of iterations $N$, and the best student policy is chosen using reward as the criterion.

**Mixture of Experts**. MOE is an ensemble model (Jacobs et al., 1991; Jordan and Xu, 1995; Yuksel et al., 2012) that consists of expert networks and a gating function. Gating function divides the input (feature) space into regions for which different experts are specialized and responsible. MOE is flexible with respect to the choice of expert models as long as they are differentiable functions of model parameters (which is not the case for DTs).

In MOE framework, probability of outputting $\mathbf{y} \in \mathbb{R}^m$ given an input $\mathbf{x} \in \mathbb{R}^n$ is given by:

$$P(\mathbf{y}|\mathbf{x}, \theta) = \sum_{i=1}^{E} P(i|\mathbf{x}, \theta_g) P(\mathbf{y}|\mathbf{x}, \theta_i) = \sum_{i=1}^{E} g_i(\mathbf{x}, \theta_g) P(\mathbf{y}|\mathbf{x}, \theta_i) \tag{1}$$

where $E$ is the number of experts, $g_i(\mathbf{x}, \theta_g)$ is the probability of choosing the expert $i$ (given input $\mathbf{x}$), $P(\mathbf{y}|\mathbf{x}, \theta_i)$ is the probability of expert $i$ producing output $\mathbf{y}$ (given input $\mathbf{x}$). Learnable parameters are $\theta = (\theta_g, \theta_e)$, where $\theta_g$ are parameters of the gating function and $\theta_e = (\theta_1, \theta_2, ..., \theta_E)$ are parameters of the experts. Gating function can be modeled using a softmax function over a set of linear models. Let $\theta_g$ consist of parameter vectors $(\theta_{g1}, \ldots, \theta_{gE})$, then the gating function can be defined as $g_i(\mathbf{x}, \theta_g) = \exp(\theta_{gi}^T \mathbf{x}) / \sum_{j=1}^{E} \exp(\theta_{gj}^T \mathbf{x})$ .

In the case of classification, an expert $i$ outputs a vector $\mathbf{y}_i$ of length $C$, where $C$ is the number of classes. Expert $i$ associates a probability to each output class $c$ (given by $\mathbf{y}_{ic}$) using the gating function. Final probability of a class $c$ is a gate weighted sum of $\mathbf{y}_{ic}$ for all experts $i \in 1, 2, ..., E$. This creates a probability vector $\mathbf{y} = (y_1, y_2, ..., y_C)$, and the output of MOE is $\arg\max_i \mathbf{y}_i$.

MOE is commonly trained using EM algorithm, where instead of direct optimization of the likelihood one performs optimization of an auxiliary function $\hat{L}$ defined in a following way. Let $z$ denote the expert chosen for instance $\mathbf{x}$. Then joint likelihood of $\mathbf{x}$ and $z$ can be considered. Since $z$ is not observed in the data, log likelihood of samples $(\mathbf{x}, z, \mathbf{y})$ cannot be computed, but instead expected log likelihood can be considered, where expectation is taken over $z$. Since the expectation has to rely on some distribution of $z$, in the iterative process, the distribution with respect to the current estimate of parameters $\theta$ is used. More precisely function $\hat{L}$ is defined by (Jordan and Xu, 1995):

$$\hat{L}(\theta, \theta^{(k)}) = \mathbb{E}_z[\log P(\mathbf{x}, z, \mathbf{y})|\mathbf{x}, \mathbf{y}, \theta^{(k)}] = \int P(z|\mathbf{x}, \mathbf{y}, \theta^{(k)}) \log P(\mathbf{x}, z, \mathbf{y}) dz \tag{2}$$

where $\theta^{(k)}$ is the estimate of parameters $\theta$ in iteration $k$. Then, for a specific sample $D = \{(\mathbf{x}_i, \mathbf{y}_i) \mid i = 1, \ldots, N\}$, the following formula can be derived (Jordan and Xu, 1995):

$$\hat{L}(\theta, \theta^{(k)}) = \sum_{i=1}^{N} \sum_{j=1}^{E} h_{ij}^{(k)} \log g_j(\mathbf{x}_i, \theta_g) + \sum_{i=1}^{N} \sum_{j=1}^{E} h_{ij}^{(k)} \log P(\mathbf{y}_i|\mathbf{x}_i, \theta_j) \tag{3}$$

where it holds

$$h_{ij}^{(k)} = \frac{g_j(\mathbf{x}_i, \theta_g^{(k)}) P(\mathbf{y}_i|\mathbf{x}_i, \theta_j^{(k)})}{\sum_{l=1}^{E} g_l(\mathbf{x}_i, \theta_g^{(k)}) P(\mathbf{y}_i|\mathbf{x}_i, \theta_l^{(k)})} \tag{4}$$

## 5 MIXTURE OF EXPERT TREES

In this section we explain the adaptation of original MoE model to mixture of decision trees, and present both training and inference algorithms.

Considering that coefficients $h_{ij}^{(k)}$ (Eq. 4) are fixed with respect to $\theta$ and that in Eq. 3 the gating part (first double sum) and each expert part depend on disjoint subsets of parameters $\theta$, training can be carried out by interchangeably optimizing the weighted log likelihood for experts (independently from one another) and optimizing the gating function with respect to the obtained experts. The training procedure for MOËT, described by Algorithm 1, is based on this observation. First, the parameters of the gating function are randomly initialized (line 2). Then the experts are trained one by one. Each expert $j$ is trained on a dataset $D_w$ of instances weighted by coefficients $h_{ij}^{(k)}$ (line 5), by applying specific DT learning algorithm (line 6) that we adapted for MoE context (described below). After the experts are trained, an optimization step is performed (line 7) in order to increase the gating part of Eq. 3. At the end, the parameters are returned (line 8).

Our tree learning procedure is as follows. Our technique modifies original MoE algorithm in that it uses DTs as experts. The fundamental difference with respect to traditional model comes from the fact that DTs do not rely on explicit and differentiable loss function which can be trained by gradient descent or Newton's methods. Instead, due to their discrete structure, they rely on a specific greedy training procedure. Therefore, the training of DTs has to be modified in order to take into account the attribution of instances to the experts given by coefficients $h_{ij}^{(k)}$, sometimes called *responsibility* of expert $j$ for instance $i$. If these responsibilities were hard, meaning that each instance is assigned to strictly one expert, they would result in partitioning the feature space into disjoint regions belonging to different experts. On the other hand, soft responsibilities are fractionally distributing each instance to different experts. The higher the responsibility of an expert $j$ for an instance $i$, the higher the influence of that instance on that expert's training. In order to formulate this principle, we consider which way the instance influences construction of a tree. First, it affects the impurity measure computed when splitting the nodes and second, it influences probability estimates in the leaves of the tree. We address these two issues next.

A commonly used impurity measure to determine splits in the tree is the Gini index. Let $U$ be a set of indices of instances assigned to the node for which the split is being computed and $D_U$ set of corresponding instances. Let categorical outcomes of $y$ be $1, \ldots, C$ and for $l = 1, \ldots, C$ denote $p_l$ fraction of assigned instances for which it holds $y = l$. More formally $p_l = \frac{\sum_{i \in U} I[y_i = l]}{|U|}$, where $I$ denotes indicator function of its argument expression and equals 1 if the expression is true. Then the Gini index $G$ of the set $D_U$ is defined by: $G(p_1, \ldots, p_C) = 1 - \sum_{l=1}^{C} p_l^2$. Considering that the assignment of instances to experts are fractional as defined by responsibility coefficients $h_{ij}^{(k)}$ (which are provided to tree fitting function as weights of instances computed in line 5 of the algorithm), this definition has to be modified in that the instances assigned to the node should not be counted, but instead, their weights should be summed. Hence, we propose the following definition:

$$\hat{p}_l = \frac{\sum_{i \in U} I[y_i = l] h_{ij}^{(k)}}{\sum_{i \in U} h_{ij}^{(k)}} \tag{5}$$

and compute the Gini index for the set $D_U$ as $G(\hat{p}_1, \ldots, \hat{p}_C)$. Similar modification can be performed for other impurity measures relying on distribution of outcomes of a categorical variable, like entropy. Note that while the instance assignments to experts are soft, instance assignments to nodes within an expert are hard, meaning sets of instances assigned to different nodes are disjoint. Probability estimate for $\mathbf{y}$ in the leaf node is usually performed by computing fractions of instances belonging to each class. In our case, the modification is the same as the one presented by Eq. 5. That way, estimates of probabilities $P(\mathbf{y}|\mathbf{x}, \theta_j^{(k)})$ needed by MoE are defined. In Algorithm 1, function $fit\_tree$ performs decision tree training using the above modifications.

We consider two ways to perform inference with respect to the obtained model. First one which we call MOËT, is performed by maximizing $P(\mathbf{y}|\mathbf{x}, \theta)$ with respect to $\mathbf{y}$ where this probability is defined by Eq. 1. The second way, which we call MOËT$_h$, performs inference as $\arg\max_{\mathbf{y}} P(\mathbf{y}|\mathbf{x}, \theta_{\arg\max_j g_j(x, \theta_g)})$, meaning that we only rely on the most probable expert.

---

**Algorithm 1** MOËT training.

---

1: **procedure** MOËT (DATA $\{(\mathbf{x}_i, \mathbf{y}_i) \mid i = 1, \ldots, N\}$, EPOCHS $N_E$, NUMBER OF EXPERTS $E$)
2: $\quad \theta_\mathbf{g} \leftarrow initialize()$
3: $\quad$ **for** $e \leftarrow 1$ to $N_E$ **do**
4: $\quad\quad$ **for** $j \leftarrow 1$ to $E$ **do**
5: $\quad\quad\quad D_w \leftarrow \left\{ \left( \mathbf{x}_i, \mathbf{y}_i, \frac{g_j(\mathbf{x}_i, \theta_g) P(\mathbf{y}_i | \mathbf{x}_i, \theta_j)}{\sum_{k=1}^E g_k(\mathbf{x}_i, \theta_g) P(\mathbf{y}_i | \mathbf{x}_i, \theta_k)} \right) \mid i = 1, \ldots, N \right\}$
6: $\quad\quad\quad \theta_j \leftarrow fit\_tree(D_w)$
7: $\quad\quad \theta_g \leftarrow \theta_g + \lambda \nabla_{\theta'} \sum_{i=1}^N \sum_{j=1}^E \left[ \frac{g_j(\mathbf{x}_i, \theta_g) P(\mathbf{y}_i | \mathbf{x}_i, \theta_j)}{\sum_{k=1}^E g_k(\mathbf{x}_i, \theta_g) P(\mathbf{y}_i | \mathbf{x}_i, \theta_k)} \log g_j(\mathbf{x}_i, \theta') \right]$
8: $\quad$ **return** $\theta_g, (\theta_1, \ldots, \theta_E)$

---

**Adaptation of MOËT to imitation learning**. We integrate MOËT model into imitation learning approach of Viper by substituting training of DT with the MOËT training procedure.

**Expressiveness**. Standard decision trees make their decisions by partitioning the feature space into regions which have borders perpendicular to coordinate axes. To approximate borders that are not perpendicular to coordinate axes very deep trees are usually necessary. MOËT$_h$ mitigates this shortcoming by exploiting hard softmax partitioning of the feature space using borders which are still hyperplanes, but need not be perpendicular to coordinate axes (see Section 3). This improves the expressiveness of the model.

**Interpretability and Verifiability**. A MOËT$_h$ model is a combination of a linear model and several decision tree models. For interpretability which is preserved in Lipton's sense of transparency, it is important that a single DT is used for each prediction (instead of weighted average). Simultability of MOËT$_h$ consisting of DT and linear models is preserved because our models are small ($2 \leq$ depth $\leq 10$) and we do not use high dimensional features (Lipton, 2016), so a person can easily simulate the model. Similarly, decomposability is preserved because simple linear models without heavily engineered features and decision trees are decomposable (Lipton, 2016) and MOËT$_h$ is a simple combination of the two. Finally, algorithmic transparency is achieved because MOËT training relies on DT training for the experts and linear model training for the gate, both of which are well understood. However, the alternating refinement of initial feature space partitioning and experts makes the procedure more complicated, so our algorithmic transparency is partially achieved. Importantly, we define a well-founded translation of MOËT$_h$ models to SMT formulas, which opens a new range of possibilities for interpreting and validating the model using automated reasoning tools. SMT formulas provide a rich means of logical reasoning, where a user can ask the solver questions such as: "On which inputs do the two models differ?", or "What is the closest input to the given input on which model makes a different prediction?", or "Are the two models equivalent?", or "Are the two models equivalent in respect to the output class C?". Answers to these and similar questions can help better understand and compare models in a rigorous way. Also note that our symbolic reasoning of the gating function and decision trees allows us to construct SMT formulas that are readily handled by off-the-shelf tools, whereas direct SMT encodings of neural networks do not scale for any reasonably sized network because of the need for non-linear arithmetic reasoning.

## 6 EVALUATION

We now compare MOËT and Viper on four OpenAI Gym environments: CartPole, Pong, Acrobot and Mountaincar. For DRL agents, we use policy gradient model in CartPole, in other environments we use a DQN (Mnih et al., 2015) (training parameters provided in appendix). The rewards obtained by the agents on CartPole, Pong, Acrobot and Mountaincar are $200.00, 21.00, -68.60$ and $-105.27$, respectively (higher reward is better). Rewards are averaged across 100 runs (250 in CartPole).

**Comparison of MOËT, MOËT$_h$, and Viper policies**. For CartPole, Acrobot, and Mountaincar environments, we train Viper DTs with maximum depths of $\{1, 2, 3, 4, 5\}$, while in the case of Pong we use maximum depths of $\{4, 6, 8, 10\}$ as the problem is more complex and requires deeper trees. For experts in MOËT policies we use the same maximum depths as in Viper (except for Pong for which we use depths 1 to 9) and we train the policies for 2 to 8 experts (in case of Pong we train for $\{2, 4, 8\}$ experts). We train all policies using 40 iterations of Viper algorithm, and choose the

Table 1: Comparison of Viper, MOËT, and MOËT$_h$ on four environments. D is effective depth of Viper and MOËT models, R is sum of rewards (return), M are mispredictions in comparison with the DRL agent, and C is a MOËT configuration (E number of experts, D depth of each expert). R and M values shown are averaged across many trials (episodes), and furthermore averaged across multiple models trained with the same configuration.

| D | Viper | | MOËT | | | MOËT$_h$ | | |
|---|---|---|---|---|---|---|---|---|
| | **R** | **M** | **R** | **M** | **C** | **R** | **M** | **C** |
| CartPole | | | | | | | | |
| 1 | **181.76** | **30.43**% | | \ | | | \ | |
| 2 | **200.00** | 16.65% | **200.00** | **0.84**% | E2:D1 | **200.00** | 0.91% | E2:D1 |
| 3 | **200.00** | 11.04% | **200.00** | 0.66% | E3:D1 | **200.00** | **0.61**% | E4:D1 |
| 4 | **200.00** | 6.87% | **200.00** | 0.92% | E3:D2 | **200.00** | **0.80**% | E4:D2 |
| 5 | **200.00** | 5.89% | **200.00** | 0.93% | E3:D3 | **200.00** | **0.87**% | E3:D3 |
| Pong | | | | | | | | |
| 4 | -6.35 | 76.50% | **6.05** | 76.48% | E8:D1 | 4.88 | **75.01**% | E4:D2 |
| 6 | 11.01 | 70.74% | 15.20 | **70.44**% | E2:D5 | **15.21** | 72.37% | E8:D3 |
| 8 | 15.96 | 68.12% | **20.24** | **64.32**% | E2:D7 | 19.13 | 66.12% | E8:D5 |
| 10 | 20.57 | 59.35% | 20.70 | 56.78% | E2:D9 | **20.73** | **54.91**% | E4:D8 |
| Acrobot | | | | | | | | |
| 2 | -86.17 | 19.83% | -82.47 | 20.50% | E2:D1 | **-81.70** | **19.18**% | E2:D1 |
| 3 | -83.40 | 19.68% | -81.68 | **17.90**% | E4:D1 | **-80.68** | 19.35% | E4:D1 |
| 4 | -82.64 | 20.17% | -79.92 | **14.89**% | E8:D1 | **-79.70** | 15.75% | E5:D1 |
| 5 | -81.99 | 17.41% | **-78.58** | **14.74**% | E7:D2 | -81.92 | 15.88% | E4:D3 |
| Mountaincar | | | | | | | | |
| 2 | -119.07 | 35.09% | **-105.53** | **21.35**% | E2:D1 | -107.15 | 22.85% | E2:D1 |
| 3 | -109.82 | 24.12% | -101.27 | 14.86% | E3:D1 | **-100.61** | **13.34**% | E3:D1 |
| 4 | -103.53 | 9.19% | **-99.67** | 8.04% | E4:D2 | -100.36 | **7.08**% | E2:D3 |
| 5 | -102.64 | 7.67% | **-100.07** | 8.84% | E6:D2 | -100.20 | **7.31**% | E8:D2 |

best performing policy in terms of rewards (and lower misprediction rate in case of equal rewards). We use two criteria to compare policies: rewards and mispredictions (number of times the student performs an action different from what a teacher would do). High reward indicates that the student learned more crucial parts of the teacher's policy, while a low misprediction rate indicates that in most cases student performs the same action as the teacher. In order to measure mispredictions, we run the student for number of runs, and compare actions it took to the actions teacher would perform.

To ensure comparable depths for evaluating Viper and MOËT models while accounting for the different number of experts in MOËT, we introduce the notion of effective depth of a MOËT model as $\lceil \log_2(E) \rceil + D$, where $E$ denotes the number of experts and $D$ denotes the depth of each expert. Table 1 compares the performance of Viper, MOËT and MOËT$_h$. The first column shows the depth of Viper decision trees and the corresponding effective depth for MOËT, rewards and mispredictions are shown in R and M columns resp. We show results of the best performing MOËT configuration for a given effective depth chosen based on average results for rewards and mispredictions, where e.g. E3:D2 denotes 3 experts with DTs of depth 2. All results shown are averaged across 10 runs[1].

For CartPole, Viper, MOËT and MOËT$_h$ all achieve perfect reward (200) with depths of 2 and greater. More interestingly, for depth 2 MOËT and MOËT$_h$ obtain significantly lower average misprediction rates of 0.84% and 0.91% respectively compared to 16.65% for Viper. Even for larger depths, the misprediction rates for MOËT and MOËT$_h$ remain significantly lower. For Pong, we observe that MOËT and MOËT$_h$ consistently outperform Viper for all depths in terms of rewards and mispredictions, whereas MOËT and MOËT$_h$ have similar performance. For Acrobot, we similarly notice that both MOËT and MOËT$_h$ achieve consistently better rewards compared to Viper for all depths. Moreover, the misprediction rates are also significantly lower for MOËT and MOËT$_h$ in majority of the cases. Finally, for Mountaincar as well, we observe that MOËT and MOËT$_h$

---

[1] except for Pong which we run for 7 times because of high computational cost.

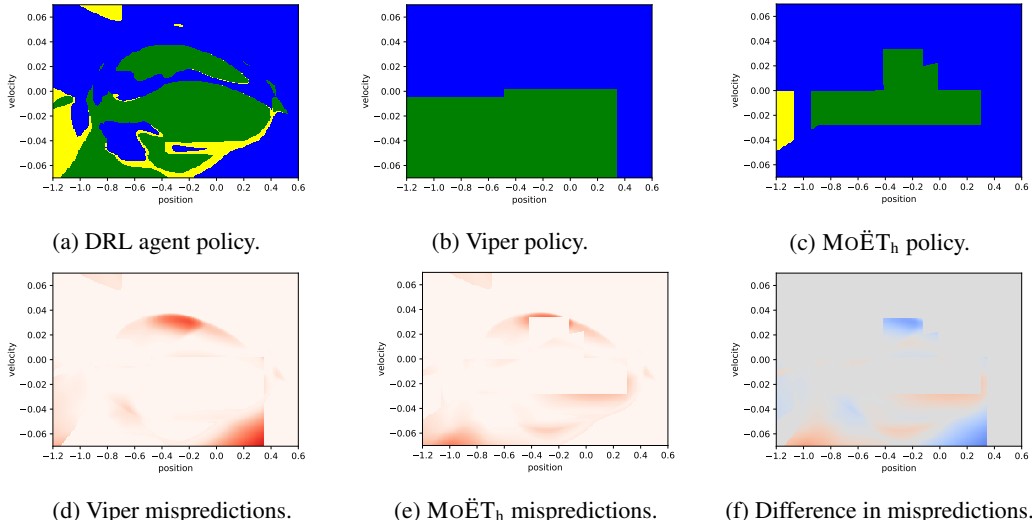

(a) DRL agent policy.    (b) Viper policy.    (c) MOËT_h policy.

(d) Viper mispredictions.  (e) MOËT_h mispredictions.  (f) Difference in mispredictions.

Figure 2: Visualization of DRL, Viper, and MOËT_h policies (up), and their differences (down) on Mountaincar environment. Upper 3 images visualize policies, where *left*, *neutral*, and *right* actions are shown in green, yellow, and blue, respectively. Lower left 2 images visualize mispredictions compared to DRL agent shown in red, where more critical mispredictions are shown with higher intensity. Finally, image in the bottom right shows difference in mispredictions between Viper and MOËT_h (blue regions is where MOËT_h is predicting correctly but Viper does not, and red the opposite, and color intensity denotes how critical the action is).

both consistently outperform Viper with significantly higher rewards and lower misprediction rates. Moreover, in both of these environments, we observe that both MOËT and MOËT_h achieve comparable reward and misprediction rates. Additional results are presented in appendix.

**Analyzing the learned Policies**. We analyze the learned student policies (Viper and MOËT_h) by visualizing their state-action space, the differences between them, and differences with the teacher policy. We use the Mountaincar environment for this analysis because of the ease of visualizing its 2-dimensional state space comprising of *car position* ($p$) and *car velocity* ($v$) features, and 3 allowed actions *left*, *right*, and *neutral*. We visualize DRL, Viper and MOËT_h policies in Figure 2, showing the actions taken in different parts of the state space (additional visualizations are in appendix).

The state space is defined by feature bounds $p \in [-1.2, 0.6]$ and $v \in [-0.07, 0.07]$, which represent sets of allowed feature values in Mountaincar. We sample the space uniformly with a resolution $200 \times 200$. The actions *left*, *neutral*, and *right* are colored in green, yellow, and blue, respectively. Recall that MOËT_h can cover regions whose borders are hyperplanes of arbitrary orientation, while Viper, i.e. DT can only cover regions whose borders are perpendicular to coordinate axes. This manifests in MOËT_h policy containing slanted borders in yellow and green regions to capture more precisely the geometry of DRL policy, while the Viper policy only contains straight borders.

Furthermore, we visualize mispredictions for Viper and MOËT_h policies. While in previous section we calculated mispredictions by using student policy for playing the game, in this analysis we visualize mispredictions across the whole state space by sampling. Note that in some states (critical states) it is more important to get the action right, while in other states choosing non-optimal action does not affect the overall score much. Viper authors make use of this observation to weight states by their importance, and they use difference between Q values of optimal and non-optimal actions as a proxy for calculating how important (critical) state is. Importance score is calculated as follows: $I(s) = \max_{a \in A} Q(s, a) - \min_{a \in A} Q(s, a)$, where $Q(s, a)$ denotes the $Q$ value of action $a$ in state $s$, and $A$ is a set of all possible actions. Using I function we weight mispredictions by their importance.

We create a vector $\mathbf{i}$ consisting of importance scores for sampled points, and normalize it to range $[0, 1]$. We also create a binary vector $\mathbf{z}$ which is 1 in the case of misprediction (student policy decision is different from DRL decision) and 0 otherwise. We visualize $\mathbf{m} = \mathbf{z} \odot \mathbf{i}$ (element-wise multiplication), where higher value indicates misprediction of higher importance and is denoted by a red color of higher intensity. Such normalized mispredictions ($\mathbf{m}$) for Viper and MOËT_h policies

are shown in Figure 2d and Figure 2e respectively. We can observe that the MOËT$_h$ policy has fewer high intensity regions leading to fewer overall mispredictions.

To provide a quantitative difference between the mispredictions of two policies, we compute $M = (\sum_j \mathbf{m}_j / \sum_j \mathbf{i}_j) \cdot 100$, which is measure in bounds $[0, 100]$ such that its value is 0 in the case of no mispredictions, and 100 in the case of all mispredictions. For the policies shown in Figure 2d and Figure 2e, we obtain $M = 15.51$ for Viper and $M = 11.78$ for MOËT$_h$ policies. We also show differences in mispredictions between Viper and MOËT$_h$ (Figure 2f), by subtracting the $\mathbf{m}$ vector of MOËT$_h$ from the $\mathbf{m}$ vector of Viper. The positive values are shown in blue and the negative values are shown in red. The higher intensity blue regions denote states where MOËT$_h$ policy gets more important action right and Viper does not (similarly vice versa for high intensity red regions).

**Translating MOËT to SMT.** We now show the translation of MOËT policy to SMT constraints for verifying policy properties. We present an example translation of MOËT policy on CartPole environment with the same property specification that was proposed for verifying Viper policies (Bastani et al., 2018). The goal in CartPole is to keep the pole upright, which can be encoded as a formula: $\psi \equiv s_0 \in S_0 \wedge \bigwedge_{t=1}^{\infty} |\phi(f_t(s_{t-1}, \pi(s_{t-1}))| \leq y_0$, where $s_i$ represents state after $i$ steps, $\phi$ is the deviation of pole from the upright position. In order to encode this formula it is necessary to encode the transition function $f_t(s, a)$ which models environment dynamics: given a state and action it returns the next state of the environment. Also, it is necessary to encode the policy function $\pi(s)$ that for a given state returns action to perform. There are two issues with verifying $\psi$: (1) infinite time horizon; and (2) the nonlinear transition function $f_t$. To solve this problem, Bastani et al. (2018) use a finite time horizon $T_{max} = 10$ and linear approximation of the dynamics and we make the same assumptions.

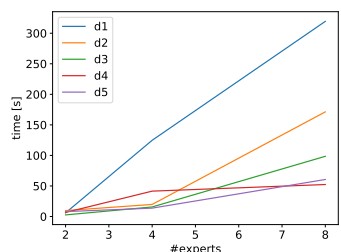

Figure 3: Verification times.

To encode $\pi(s)$ we need to translate both the gating function and DT experts to logical formulas. Since the gating function in MOËT$_h$ uses exponential function, it is difficult to encode the function directly in Z3 as SMT solvers do not have efficient decision procedures to solve non-linear arithmetic. The direct encoding of exponentiation therefore leads to prohibitively complex Z3 formulas. We exploit the following simplification of gating function that is sound when hard prediction is used:

$$e = \arg\max_i \left( \frac{\exp(\theta_{gi}^T \mathbf{x})}{\sum_{j=1}^{E} \exp(\theta_{gj}^T \mathbf{x})} \right) = \arg\max_i (\exp(\theta_{gi}^T \mathbf{x})) = \arg\max_i (\theta_{gi}^T \mathbf{x})$$

First simplification is possible since the denominators for gatings of all experts are same, and second simplification is due to the monotonicity of the exponential function. For encoding DTs we use the same encoding as in Viper. To verify that $\psi$ holds we need to show that $\neg\psi$ is unsatisfiable. We run the verification with our MOËT$_h$ policies and show that $\neg\psi$ is indeed unsatisfiable.

To better understand the scalability of our verification procedure, we report on the verification times needed to verify policies for different number of experts and different expert depths in Figure 3. We observe that while MOËT$_h$ policies with 2 experts take from 2.6s to 8s for verification, the verification times for 8 experts can go up to as much as 319s. This directly corresponds to the complexity of the logical formula obtained with an increase in the number of experts.

# 7 CONCLUSION

We introduced MOËT, a technique based on MOE with expert decision trees and presented a learning algorithm to train MOËT models. We then used MOËT models for interpreting DRL agent policies, where different local DTs specialize on different regions of input space and are combined into a global policy using a gating function. We showed that MOËT models lead to smaller, more faithful and performant representation of DRL agents compared to previous state-of-the-art approaches like Viper while still maintaining interpretability and verifiability.

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

---

**Algorithm 2** Viper training (Bastani et al., 2018)

---

1: **procedure** VIPER (MDP $e$, TEACHER $\pi_t$, Q-FUNCTION $Q^{\pi_t}$, ITERATIONS $N$)
2:     Initialize dataset and student: $D \leftarrow \emptyset, \pi_{s_0} \leftarrow \pi_t$
3:     **for** $i \leftarrow 1$ to $N$ **do**
4:         Sample trajectories and aggregate: $D \leftarrow D \cup \{(s, \pi_t(s)) \sim d^{\pi_{s_{i-1}}}(e)\}$
5:         Sample dataset using Q values: $D_s \leftarrow \{(s, a) \in I \sim D\}$
6:         Train decision tree: $\pi_{s_i} \leftarrow fit\_tree(D_s)$
7:     **return** Best policy $\pi_s \in \{\pi_{s_1}, ..., \pi_{s_N}\}$.

---

## A  VIPER ALGORITHM

Viper algorithm is shown in Algorithm 2.

## B  DRL AGENT TRAINING PARAMETERS

Here we present parameters we used to train DRL agents for different environments. For CartPole, we use policy gradient model as used in Viper. While we use the same model, we had to retrain it from scratch as the trained Viper agent was not available. For Pong, we use a deep Q-network (DQN) network (Mnih et al., 2015), and we use the same model as in Viper, which originates from OpenAI baselines (OpenAI Baselines). For Acrobot and Mountaincar, we implement our own version of dueling DQN network following (Wang et al., 2015). We use 3 hidden layers with 15 neurons in each layer. We set the learning rate to $0.001$, batch size to $30$, step size to $10000$ and number of epochs to $80000$. We checkpoint a model every $5000$ steps and pick the best performing one in terms of achieved reward.

## C  ENVIRONMENTS

In this section we provide a brief description of environments we used in our experiments. We used four environments from OpenAI Gym: CartPole, Pong, Acrobot and Mountaincar.

### C.1  CARTPOLE

This environment consists of a cart and a rigid pole hinged to the cart, based on the system presented by Barto et al. (Barto et al., 1983). At the beginning pole is upright, and the goal is to prevent it from falling over. Cart is allowed to move horizontally within predefined bounds, and controller chooses to apply either *left* or *right* force to the cart. State is defined with four variables: $x$ (cart position), $\dot{x}$ (cart velocity), $\theta$ (pole angle), and $\dot{\theta}$ (pole angular velocity). Game is terminated when the absolute value of pole angle exceeds $12°$, cart position is more than $2.4$ units away from the center, or after $200$ successful steps; whichever comes first. In each step reward of $+1$ is given, and the game is considered solved when the average reward is over $195$ in over $100$ consecutive trials.

### C.2  PONG

This is a classical Atari game of table tennis with two players. Minimum possible score is $-21$ and maximum is $21$.

### C.3  ACROBOT

This environment is analogous to a gymnast swinging on a horizontal bar, and consists of a two links and two joins, where the joint between the links is actuated. The environment is based on the system presented by Sutton (Sutton, 1996). Initially both links are pointing downwards, and the goal is to swing the end-point (feet) above the bar for at least the length of one link. The state consists of six variables, four variables consisting of sin and cos values of the joint angles, and two variables for angular velocities of the joints. The action is either applying *negative*, *neutral*, or *positive* torque on the joint. At each time step reward of $-1$ is received, and episode is terminated upon successful

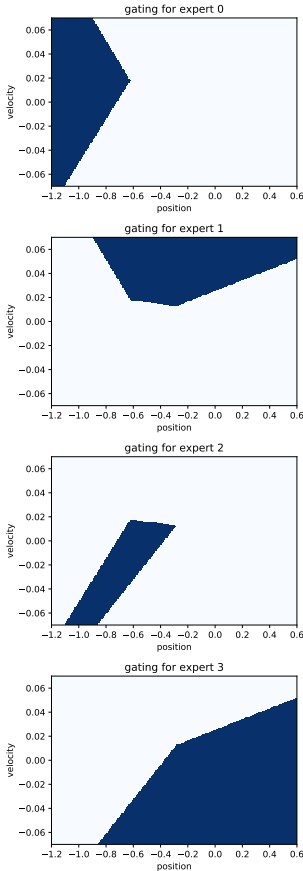

Figure 4: Visualization of gating function for different experts.

reaching the height, or after 200 steps, whichever comes first. Acrobot is an unsolved environment in that there is no reward limit under which is considered solved, but the goal is to achieve high reward.

### C.4 MOUNTAINCAR

This environment consists of a car positioned between two hills, with a goal of reaching the hill in front of the car. The environment is based on the system presented by Moore (Moore, 1990). Car can move in a one-dimensional track, but does not have enough power to reach the hill in one go, thus it needs to build momentum going back and forth to finally reach the hill. Controller can choose *left*, *right* or *neutral* action to apply left, right or no force to the car. State is defined by two variables, describing car position and car velocity. In each step reward of $-1$ is received, and episode is terminated upon reaching the hill, or after 200 steps, whichever comes first. The game is considered solved if average reward over 100 consecutive trials is no less than $-110$.

## D    ADDITIONAL VISUALIZATIONS

In this section we provide visualization of a gating function. Figure 4 shows how gating function partitions the state space for which different experts specialize. Gatings of MoËT$_h$ policy with 4 experts and depth 1 are shown.

Table 2: Comparison of Viper, MOËT, and MOËT$_h$ on four environments. D is effective depth of Viper and MOËT models, R is sum of rewards (return), M are mispredictions in comparison with the DRL agent. Results shown are averaged across multiple trained models for Viper and MOËT. In addition, MOËT results are averaged across multiple configurations that have the same effective depth.

| | **Viper** | | **MOËT** | | **MOËT$_h$** | |
|---|---|---|---|---|---|---|
| **D** | **R** | **M** | **R** | **M** | **R** | **M** |
| | | | CartPole | | | |
| 1 | **181.76** | **30.43**% | | % | | % |
| 2 | **200.00** | 16.65% | **200.00** | **0.84**% | **200.00** | 0.91% |
| 3 | **200.00** | 11.04% | **200.00** | 0.87% | **200.00** | **0.84**% |
| 4 | **200.00** | 6.87% | **200.00** | 1.01% | **200.00** | **0.96**% |
| 5 | **200.00** | 5.89% | **200.00** | **1.16**% | **200.00** | 1.17% |
| | | | Pong | | | |
| 4 | -6.35 | 76.50% | 3.03 | **74.48**% | **3.08** | 75.18% |
| 6 | 11.01 | **70.74**% | **14.92** | 71.49% | 12.66 | 72.59% |
| 8 | 15.96 | 68.12% | **18.50** | **64.74**% | 17.99 | 65.94% |
| 10 | 20.57 | 59.35% | **20.66** | 55.67% | 20.29 | **52.81**% |
| | | | Acrobot | | | |
| 2 | -86.17 | 19.83% | -82.47 | 20.50% | **-81.70** | **19.18**% |
| 3 | -83.40 | 19.68% | -82.45 | **19.34**% | **-81.79** | 19.52% |
| 4 | -82.64 | 20.17% | -82.36 | 17.61% | **-81.52** | **17.37**% |
| 5 | **-81.99** | 17.41% | -82.44 | **16.16**% | -83.57 | 17.44% |
| | | | Mountaincar | | | |
| 2 | -119.07 | 35.09% | **-105.53** | **21.35**% | -107.15 | 22.85% |
| 3 | -109.82 | 24.12% | **-101.57** | 12.62% | -101.68 | **12.42**% |
| 4 | -103.53 | **9.19**% | **-100.83** | 10.12% | -100.96 | 10.03% |
| 5 | -102.64 | **7.67**% | -101.06 | 8.03% | **-100.69** | 7.73% |

Table 3: Comparison of the best performing models for DRL, MOËT and MOËT$_h$.

| | **DRL** | **MOËT** | **MOËT$_h$** |
|---|---|---|---|
| CartPole | 200 | 200 | 200 |
| Pong | 21 | 21 | 21 |
| Acrobot | -68.60 | -73.64 | -73.31 |
| Mountaincar | -105.27 | -97.57 | -97.59 |

# E ADDITIONAL TABLES

Table 2 shows results similar to Table 1, but here in addition to averaging results across multiple trained models, it averages results across multiple MOËT configurations that have the same effective depth.

Table 3 shows the results of best performing DRL, MOËT and MOËT$_h$ models on the evaluation subjects.

Table 4: cartpole ViperPlus Evaluation

| D | R | M |
|---|---|---|
| 1 | 181.76 | 30.43% |
| 2 | 200.00 | 16.65% |
| 3 | 200.00 | 11.04% |
| 4 | 200.00 | 6.87% |
| 5 | 200.00 | 5.89% |

Table 5: cartpole MOE Evaluation

| E | D | R | M |
|---|---|---|---|
| 2 | 1 | 200.00 | 0.84% |
| 2 | 2 | 200.00 | 1.16% |
| 2 | 3 | 200.00 | 1.04% |
| 2 | 4 | 200.00 | 1.58% |
| 2 | 5 | 200.00 | 2.44% |
| 3 | 1 | 200.00 | 0.66% |
| 3 | 2 | 200.00 | 0.92% |
| 3 | 3 | 200.00 | 0.93% |
| 3 | 4 | 200.00 | 1.37% |
| 3 | 5 | 200.00 | 2.37% |
| 4 | 1 | 200.00 | 0.80% |
| 4 | 2 | 200.00 | 0.97% |
| 4 | 3 | 200.00 | 0.96% |
| 4 | 4 | 200.00 | 1.53% |
| 4 | 5 | 199.96 | 2.71% |
| 5 | 1 | 200.00 | 0.92% |
| 5 | 2 | 200.00 | 1.02% |
| 5 | 3 | 200.00 | 1.26% |
| 5 | 4 | 200.00 | 1.97% |
| 5 | 5 | 200.00 | 3.01% |
| 6 | 1 | 200.00 | 0.99% |
| 6 | 2 | 200.00 | 1.27% |
| 6 | 3 | 200.00 | 1.17% |
| 6 | 4 | 200.00 | 1.97% |
| 6 | 5 | 200.00 | 2.68% |
| 7 | 1 | 200.00 | 0.93% |
| 7 | 2 | 200.00 | 1.07% |
| 7 | 3 | 200.00 | 1.64% |
| 7 | 4 | 200.00 | 2.67% |
| 7 | 5 | 200.00 | 2.91% |
| 8 | 1 | 200.00 | 1.29% |
| 8 | 2 | 200.00 | 1.26% |
| 8 | 3 | 200.00 | 1.57% |
| 8 | 4 | 200.00 | 2.23% |
| 8 | 5 | 200.00 | 3.27% |

# F    ABLATION RESULTS

In this section we show results for all DT depths and numbers of experts used for training Viper and MOËT policies. Average mispredictions and rewards are shown for all configurations. Tables 4,5,6 show results for CartPole. Tables 7,8,9 show results for Pong. Tables 10,11,12 show results for Acrobot. Tables 13,14,15 show results for Mountaincar.

Table 6: cartpole MOEHard Evaluation

| E | D | R | M |
|---|---|---|---|
| 2 | 1 | 200.00 | 0.91% |
| 2 | 2 | 200.00 | 0.98% |
| 2 | 3 | 200.00 | 1.14% |
| 2 | 4 | 200.00 | 1.34% |
| 2 | 5 | 200.00 | 2.31% |
| 3 | 1 | 200.00 | 0.93% |
| 3 | 2 | 200.00 | 0.95% |
| 3 | 3 | 200.00 | 0.87% |
| 3 | 4 | 200.00 | 1.42% |
| 3 | 5 | 200.00 | 2.48% |
| 4 | 1 | 200.00 | 0.61% |
| 4 | 2 | 200.00 | 0.80% |
| 4 | 3 | 200.00 | 1.15% |
| 4 | 4 | 200.00 | 1.86% |
| 4 | 5 | 200.00 | 2.58% |
| 5 | 1 | 200.00 | 0.92% |
| 5 | 2 | 200.00 | 0.98% |
| 5 | 3 | 200.00 | 1.31% |
| 5 | 4 | 200.00 | 1.95% |
| 5 | 5 | 200.00 | 3.17% |
| 6 | 1 | 200.00 | 0.87% |
| 6 | 2 | 200.00 | 1.02% |
| 6 | 3 | 200.00 | 1.40% |
| 6 | 4 | 200.00 | 2.10% |
| 6 | 5 | 200.00 | 2.95% |
| 7 | 1 | 200.00 | 0.90% |
| 7 | 2 | 200.00 | 1.59% |
| 7 | 3 | 200.00 | 1.36% |
| 7 | 4 | 200.00 | 2.33% |
| 7 | 5 | 200.00 | 3.30% |
| 8 | 1 | 200.00 | 1.12% |
| 8 | 2 | 200.00 | 1.26% |
| 8 | 3 | 200.00 | 1.70% |
| 8 | 4 | 200.00 | 2.57% |
| 8 | 5 | 200.00 | 2.84% |

Table 7: pong ViperPlus Evaluation

| D | R | M |
|---|---|---|
| 4 | -6.35 | 76.50% |
| 6 | 11.01 | 70.74% |
| 8 | 15.96 | 68.12% |
| 10 | 20.57 | 59.35% |

Table 8: pong MOE Evaluation

| E | D | R | M |
|---|---|---|---|
| 2 | 3 | 0.09 | 74.09% |
| 2 | 5 | 15.20 | 70.44% |
| 2 | 7 | 20.24 | 64.32% |
| 2 | 9 | 20.70 | 56.78% |
| 4 | 2 | 2.95 | 72.88% |
| 4 | 4 | 14.70 | 70.27% |
| 4 | 6 | 18.01 | 65.56% |
| 4 | 8 | 20.65 | 56.72% |
| 8 | 1 | 6.05 | 76.48% |
| 8 | 3 | 14.84 | 73.77% |
| 8 | 5 | 17.27 | 64.35% |
| 8 | 7 | 20.62 | 53.52% |

Table 9: pong MOEHard Evaluation

| E | D | R | M |
|---|---|---|---|
| 2 | 3 | 4.60 | 77.23% |
| 2 | 5 | 7.64 | 74.29% |
| 2 | 7 | 16.80 | 68.04% |
| 2 | 9 | 19.51 | 56.33% |
| 4 | 2 | 4.88 | 75.01% |
| 4 | 4 | 15.13 | 71.10% |
| 4 | 6 | 18.04 | 63.66% |
| 4 | 8 | 20.73 | 54.91% |
| 8 | 1 | -0.23 | 73.31% |
| 8 | 3 | 15.21 | 72.37% |
| 8 | 5 | 19.13 | 66.12% |
| 8 | 7 | 20.63 | 47.19% |

Table 10: acrobot ViperPlus Evaluation

| D | R | M |
|---|---|---|
| 2 | -86.17 | 19.83% |
| 3 | -83.40 | 19.68% |
| 4 | -82.64 | 20.17% |
| 5 | -81.99 | 17.41% |

Table 11: acrobot MOE Evaluation

| E | D | R | M |
|---|---|---|---|
| 2 | 1 | -82.47 | 20.50% |
| 2 | 2 | -83.51 | 20.63% |
| 2 | 3 | -85.61 | 19.45% |
| 2 | 4 | -84.22 | 16.97% |
| 2 | 5 | -82.22 | 15.49% |
| 3 | 1 | -82.17 | 19.48% |
| 3 | 2 | -82.34 | 19.79% |
| 3 | 3 | -83.69 | 17.47% |
| 3 | 4 | -86.89 | 17.93% |
| 3 | 5 | -82.28 | 15.13% |
| 4 | 1 | -81.68 | 17.90% |
| 4 | 2 | -82.62 | 17.92% |
| 4 | 3 | -84.28 | 16.91% |
| 4 | 4 | -84.36 | 16.34% |
| 4 | 5 | -82.86 | 14.28% |
| 5 | 1 | -83.70 | 17.87% |
| 5 | 2 | -83.40 | 17.75% |
| 5 | 3 | -83.12 | 16.84% |
| 5 | 4 | -85.50 | 15.53% |
| 5 | 5 | -84.53 | 14.67% |
| 6 | 1 | -81.22 | 17.49% |
| 6 | 2 | -82.74 | 15.07% |
| 6 | 3 | -84.07 | 16.23% |
| 6 | 4 | -82.64 | 13.52% |
| 6 | 5 | -84.45 | 13.75% |
| 7 | 1 | -81.10 | 15.88% |
| 7 | 2 | -78.58 | 14.74% |
| 7 | 3 | -84.00 | 16.07% |
| 7 | 4 | -81.84 | 13.89% |
| 7 | 5 | -84.96 | 13.19% |
| 8 | 1 | -79.92 | 14.89% |
| 8 | 2 | -80.19 | 14.23% |
| 8 | 3 | -83.27 | 15.39% |
| 8 | 4 | -81.61 | 12.77% |
| 8 | 5 | -82.58 | 13.28% |

Table 12: acrobot MOEHard Evaluation

| E | D | R | M |
|---|---|---|---|
| 2 | 1 | -81.70 | 19.18% |
| 2 | 2 | -83.61 | 19.68% |
| 2 | 3 | -82.01 | 19.42% |
| 2 | 4 | -83.86 | 19.37% |
| 2 | 5 | -85.60 | 18.11% |
| 3 | 1 | -81.08 | 19.52% |
| 3 | 2 | -82.71 | 19.77% |
| 3 | 3 | -86.51 | 20.72% |
| 3 | 4 | -83.92 | 17.50% |
| 3 | 5 | -82.01 | 14.70% |
| 4 | 1 | -80.68 | 19.35% |
| 4 | 2 | -81.19 | 18.72% |
| 4 | 3 | -81.92 | 15.88% |
| 4 | 4 | -81.86 | 14.68% |
| 4 | 5 | -84.64 | 15.66% |
| 5 | 1 | -79.70 | 15.75% |
| 5 | 2 | -84.37 | 18.84% |
| 5 | 3 | -85.08 | 17.87% |
| 5 | 4 | -82.42 | 14.58% |
| 5 | 5 | -84.11 | 14.83% |
| 6 | 1 | -81.51 | 16.88% |
| 6 | 2 | -83.27 | 16.37% |
| 6 | 3 | -85.06 | 15.77% |
| 6 | 4 | -83.52 | 13.84% |
| 6 | 5 | -82.11 | 13.81% |
| 7 | 1 | -83.57 | 16.61% |
| 7 | 2 | -82.47 | 15.50% |
| 7 | 3 | -85.09 | 17.11% |
| 7 | 4 | -81.71 | 12.82% |
| 7 | 5 | -82.86 | 13.24% |
| 8 | 1 | -79.96 | 14.47% |
| 8 | 2 | -82.56 | 15.41% |
| 8 | 3 | -82.39 | 15.31% |
| 8 | 4 | -82.97 | 13.99% |
| 8 | 5 | -84.50 | 13.17% |

Table 13: mountaincar ViperPlus Evaluation

| D | R | M |
|---|---|---|
| 2 | -119.07 | 35.09% |
| 3 | -109.82 | 24.12% |
| 4 | -103.53 | 9.19% |
| 5 | -102.64 | 7.67% |

Table 14: mountaincar MOE Evaluation

| E | D | R | M |
|---|---|---|---|
| 2 | 1 | -105.53 | 21.35% |
| 2 | 2 | -101.58 | 9.97% |
| 2 | 3 | -100.35 | 7.41% |
| 2 | 4 | -103.81 | 10.06% |
| 2 | 5 | -104.60 | 6.92% |
| 3 | 1 | -101.27 | 14.86% |
| 3 | 2 | -100.51 | 7.95% |
| 3 | 3 | -101.27 | 8.23% |
| 3 | 4 | -103.95 | 8.91% |
| 3 | 5 | -104.00 | 6.31% |
| 4 | 1 | -101.87 | 13.03% |
| 4 | 2 | -99.67 | 8.04% |
| 4 | 3 | -100.65 | 7.39% |
| 4 | 4 | -105.02 | 7.52% |
| 4 | 5 | -103.49 | 6.39% |
| 5 | 1 | -101.58 | 11.70% |
| 5 | 2 | -100.50 | 6.63% |
| 5 | 3 | -100.78 | 7.24% |
| 5 | 4 | -103.93 | 8.36% |
| 5 | 5 | -105.33 | 6.96% |
| 6 | 1 | -101.01 | 10.73% |
| 6 | 2 | -100.07 | 8.84% |
| 6 | 3 | -101.45 | 7.50% |
| 6 | 4 | -104.99 | 7.83% |
| 6 | 5 | -104.60 | 6.92% |
| 7 | 1 | -100.57 | 11.72% |
| 7 | 2 | -100.79 | 7.69% |
| 7 | 3 | -102.05 | 8.83% |
| 7 | 4 | -104.44 | 7.21% |
| 7 | 5 | -103.39 | 4.82% |
| 8 | 1 | -102.15 | 13.32% |
| 8 | 2 | -100.36 | 7.40% |
| 8 | 3 | -102.06 | 7.00% |
| 8 | 4 | -105.07 | 6.48% |
| 8 | 5 | -104.91 | 6.34% |

Table 15: mountaincar MOEHard Evaluation

| E | D | R | M |
|---|---|---|---|
| 2 | 1 | -107.15 | 22.85% |
| 2 | 2 | -102.37 | 12.27% |
| 2 | 3 | -100.36 | 7.08% |
| 2 | 4 | -101.33 | 7.97% |
| 2 | 5 | -104.40 | 7.10% |
| 3 | 1 | -100.61 | 13.34% |
| 3 | 2 | -100.79 | 8.10% |
| 3 | 3 | -100.57 | 6.96% |
| 3 | 4 | -102.71 | 7.61% |
| 3 | 5 | -103.97 | 5.90% |
| 4 | 1 | -102.06 | 11.65% |
| 4 | 2 | -100.36 | 9.77% |
| 4 | 3 | -101.17 | 7.89% |
| 4 | 4 | -104.29 | 6.84% |
| 4 | 5 | -104.21 | 5.65% |
| 5 | 1 | -100.86 | 11.05% |
| 5 | 2 | -100.42 | 7.57% |
| 5 | 3 | -101.27 | 7.59% |
| 5 | 4 | -104.00 | 5.94% |
| 5 | 5 | -104.16 | 4.70% |
| 6 | 1 | -100.68 | 9.62% |
| 6 | 2 | -100.60 | 8.56% |
| 6 | 3 | -100.68 | 7.14% |
| 6 | 4 | -104.92 | 6.97% |
| 6 | 5 | -104.40 | 6.05% |
| 7 | 1 | -100.59 | 11.85% |
| 7 | 2 | -100.52 | 7.84% |
| 7 | 3 | -101.67 | 7.06% |
| 7 | 4 | -102.81 | 5.27% |
| 7 | 5 | -105.14 | 8.18% |
| 8 | 1 | -103.09 | 12.71% |
| 8 | 2 | -100.20 | 7.31% |
| 8 | 3 | -101.41 | 6.69% |
| 8 | 4 | -104.67 | 5.73% |
| 8 | 5 | -105.80 | 4.69% |

