# OpenReview forum: "MoET: Interpretable and Verifiable Reinforcement Learning via Mixture of Expert Trees"
_ICLR.cc/2020/Conference — Reject_

### Official Review · AnonReviewer4 · 2019-10-21
**Official Blind Review #4**

**Rating:** 6

**Review:**

The paper suggests learning decision trees to learn policies that can be verified efficiently. The decision trees are learnt by imitation learning (DAGGER) and are guided by a DNN policy (called the oracle) and its Q-function. The paper is an extension of VIPER to mixture of Experts.

The paper does two contributions:
- the modification of Viper to learn a mixture of decision tree policies that better mimics the DRL agent, and
- the fact that the model based on Mixture of Experts is still interpretable.

Form my understanding, it is known that mixture of experts can be more accurate than a single decision tree. I'm unsure about the importance of verifying experimentally that it is also the case in the context of DAGGER. Could you please expand on why using a mixture of experts in the context of DAGGER is challenging? I think that would be an important addition to the paper. That being said, the experimental results are of interest as they bring interpretability and verifiable RL to more challenging environments.

Minor comments:
- The notation "EM" that appears in the fourth paragraph of the introduction is not defined previously.
- There a few typos: e.g., "there do not exist"
- How many seeds are used for the "average results (e.g. Appendix E)?

**Experience Assessment:**

I have published one or two papers in this area.

**Review Assessment: Checking Correctness Of Derivations And Theory:**

I assessed the sensibility of the derivations and theory.

**Review Assessment: Checking Correctness Of Experiments:**

I assessed the sensibility of the experiments.

**Review Assessment: Thoroughness In Paper Reading:**

I read the paper thoroughly.

---

> ### Author Response · Authors · 2019-11-09
> **Response to Official Blind Review #4**
>
> Thank you for the review and helpful comments. Below we provide clarifications and comments.
>
> > it is known that mixture of experts can be more accurate than a single decision tree. I'm unsure about the importance of verifying experimentally that it is also the case in the context of DAGGER.
>
> Indeed, it is expected that the mixture of experts model performs better than a single decision tree—this in fact motivates our work. However, there are key challenges in embodying the mixture into a verifiable and interpretable model, which our work shows how to address. To support interpretability we formulate a novel training procedure with decision tree experts. In addition, we show how mixture of experts with hard thresholding can be translated down to SMT to enable verification. To the best of our knowledge, MOET is the first technique to show how to combine decision trees with a mixture of experts, and it is applicable even outside of the RL setting.
>
> > Could you please expand on why using a mixture of experts in the context of DAGGER is challenging? I think that would be an important addition to the paper.
>
> Off-the-shelf mixture of experts models can directly be used with DAGGER without any problem. However, as we pointed out earlier, there was no previous mixture of experts model using the decision tree experts, because of the challenge that decision trees are not differentiable. We defined such a combination in order to construct an interpretable model. More importantly, this combination allowed us to define a translation algorithm that translates a MOET model to an SMT formula, which allows reasoning in logical space. Once translated to SMT, we can verify properties of the model, check for equivalence between two models, verify that agent never loses (assuming we have model of the environment), etc. We will add this discussion to the paper.
>
> > - The notation "EM" that appears in the fourth paragraph of the introduction is not defined previously.
> > - There a few typos: e.g., "there do not exist"
>
> Thanks for pointing this out. We will correct the mistakes in the paper revision.
>
> > - How many seeds are used for the "average results (e.g. Appendix E)?
>
> We train 10 models (7 in a case of Pong) for each configuration (number of experts/depth), and average model performance across 100 episodes (250 in a case of Cartpole). In summary, the results shown are averaged across many trials (episodes) and across different models trained with the same parameters.
>
> We hope our response answers your questions, but please let us know in case of any additional questions or clarifications.

---

### Official Review · AnonReviewer3 · 2019-10-22
**Official Blind Review #3**

**Rating:** 3

**Review:**

The paper proposes an extension to the Viper[1] method for interpreting and verifying deep RL policies by learning a mixture of decision trees to mimic the originally learned policy. The proposed approach can imitate the deep policy better compared with Viper while preserving verifiability. Empirically the proposed method demonstrates improvement in terms of cumulative reward and misprediction rate over Viper in four benchmark tasks.

I tend to reject the paper in its current form because (1) the idea of using mixture of trees to do policy extraction is somewhat incremental; and (2) experiments do not consistently show significant performance gain over the existing approach.

=============================================================================================

Novelty and significance

The paper addresses an important problem in RL, trying to extract an interpretable and verifiable policy from a deep RL model. The novel aspect of the paper is employing a mixture of expert trees model instead of a single decision tree in Viper. The proposed method is somewhat incremental in the sense that it is equivalent to replacing the first layer of the hard decision tree with a soft decision layer, which enables learning a non-axis-align decision boundaries in the first layer. This is more like a small modification of the original method than a significant contribution.

Moreover, in the experiments, the proposed method does not consistently show significant improvements over Viper. Of course one can expect some improvements over Viper because the proposed method is using more flexible model by employing a soft and non-axis-align decision layer. Besides, for Viper the comparison is not fair because it is comparing with the best performing mixture of expert trees model among many candidate structures. I would like to see how Viper compares with the average performance of different structures.

=============================================================================================

Other comments

The organization of this paper is good and the paper is easy to read. However the authors can improve the tables and figures by providing more descriptions to be more self-contained. For example, I don't know what the column labels in table 1 (D/R/M/C) are referring to by solely looking at the table. And their actual meanings are really hard to find in the text. And for figure 2, the authors should point out the meaning of colors inside the illustrations.






[1] Bastani, Osbert, Yewen Pu, and Armando Solar-Lezama. "Verifiable reinforcement learning via policy extraction." Advances in Neural Information Processing Systems. 2018.

Update after rebuttal: score unchanged


**Experience Assessment:**

I have read many papers in this area.

**Review Assessment: Checking Correctness Of Derivations And Theory:**

I assessed the sensibility of the derivations and theory.

**Review Assessment: Checking Correctness Of Experiments:**

I carefully checked the experiments.

**Review Assessment: Thoroughness In Paper Reading:**

I read the paper thoroughly.

---

> ### Author Response · Authors · 2019-11-09
> **Response to Official Blind Review #3**
>
> Thanks for the review and feedback. Below we provide clarifications and comments on the questions and concerns raised.
>
> > The proposed method is somewhat incremental in the sense that it is equivalent to replacing the first layer of the hard decision tree with a soft decision layer, which enables learning a non-axis-align decision boundaries in the first layer. This is more like a small modification of the original method than a significant contribution.
>
> Although the models are related, they cannot be trained in a similar way, therefore the method is significantly modified. Traditional mixture of experts model is defined for differentiable experts. Unfortunately, decision trees are not differentiable, and in this paper we present an algorithm that adapts mixture of experts training to support decision tree experts. To the best of our knowledge, MOET is the first technique to show how to combine decision trees with a mixture of experts, and it is applicable even outside of the RL setting.
>
> > the proposed method does not consistently show significant improvements over Viper. Of course one can expect some improvements over Viper because the proposed method is using more flexible model by employing a soft and non-axis-align decision layer.
>
> MOET generalizes Viper, and the experiments show that MOET performs consistently better than Viper. For example, the motivation section illustrates a case where MOET is significantly better than Viper. Indeed, we do not expect that to always be the case.
>
> > Besides, for Viper the comparison is not fair because it is comparing with the best performing mixture of expert trees model among many candidate structures. I would like to see how Viper compares with the average performance of different structures.
>
> Indeed, for a given depth, MOET allows multiple ways to configure the model by alternating number of individual decision trees and their depth. For fair comparison, for both Viper and MOET we train 10 different models for each configuration (structure), and average results across them. Only after averaging we choose the best performing MOET model for a given effective depth and compare it to Viper.
>
> We have now also computed the average MOET performance across different configurations and we show them below. As can be observed MOET outperforms Viper in majority of cases even when averaged across different configurations. We will also add this in the revision.
>
> ﻿                 Viper                       MOET                   MOET_h
> D           R           M              R             M            R               M
>                                          Cartpole
> 1      181.76    30.43%
> 2      200.00    16.65%    200.00      0.84%    200.00       0.91%
> 3      200.00    11.04%    200.00      0.87%    200.00       0.84%
> 4      200.00      6.87%    200.00      1.01%    200.00       0.96%
> 5      200.00      5.89%    200.00      1.16%    200.00       1.17%
>                                            Pong
> 4         -6.35    76.50%        3.03    74.48%        3.08     75.18%
> 6        11.01    70.74%      14.92    71.49%      12.66     72.59%
> 8        15.96    68.12%      18.50    64.74%      17.99     65.94%
> 10      20.57    59.35%      20.66    55.67%      20.29     52.81%
>                                          Acrobat
> 2       -86.17    19.83%     -82.47    20.50%    -81.70     19.18%
> 3       -83.40    19.68%     -82.45    19.34%    -81.79     19.52%
> 4       -82.64    20.17%     -82.36    17.61%    -81.52     17.37%
> 5       -81.99    17.41%     -82.44    16.16%    -83.57     17.44%
>                                      Mountaincar
> 2     -119.07    35.09%   -105.53    21.35%  -107.15     22.85%
> 3     -109.82    24.12%   -101.57    12.62%  -101.68     12.42%
> 4     -103.53      9.19%   -100.83    10.12%  -100.96     10.03%
> 5     -102.64      7.67%   -101.06      8.03%  -100.69       7.73%
>
> R - rewards, M - mispredictions, D - depth (effective depth in a case of MOET)
>
>  > However the authors can improve the tables and figures by providing more descriptions to be more self-contained. For example, I don't know what the column labels in table 1 (D/R/M/C) are referring to by solely looking at the table. And their actual meanings are really hard to find in the text. And for figure 2, the authors should point out the meaning of colors inside the illustrations.
>
> Thanks for the suggestion. We will make these changes in our revision: Table 1 to be more self contained, and Figure 2 to describe meanings of different colors.
>
> We hope that we have clarified the concerns about novelty of MOET — the first technique to show how to combine decision trees (non-differentiable experts) with a mixture of experts, and that MOET consistently performs better than Viper while still maintaining interpretability and verifiability. Please let us know if there might be any other questions or comments.

---

> > ### Author Response · Authors · 2019-11-13
> > **Feedback to Response**
> >
> > Dear reviewer,
> >
> > Please let us know if our response helped address your concerns regarding the novelty of our mixture of (non-differentiable) expert decision trees, improvements over Viper (decision trees) both theoretically and empirically, and fairness of comparison with Viper. Please let us know if there might be any additional questions or clarifications needed for possible reconsideration.
> >
> > Thanks you,
> > Authors

---

> > > ### Comment · AnonReviewer3 · 2019-11-15
> > > **Thanks for the response**
> > >
> > > Thanks for the author response! However, the response does not convince me that the contribution of the paper is significant enough to be accepted. I will keep my rating unchanged.

---

### Official Review · AnonReviewer2 · 2019-10-22
**Official Blind Review #2**

**Rating:** 6

**Review:**

The paper proposes a method (MOET) to distillate a reinforcement learning policy  represented by a deep neural network into an ensemble of decision trees. The main objective of this procedure is to obtain an "interpretable" and verifiable policy while maintaining the performance of the policy. The authors build over the previously published algorithm Viper (Bastani et al, 2018), which distillates deep policies into a single decision tree using the DAGGER procedures, i.e. alternation of imitation learning of an expert policy and of additional data-sampling from the newly learned policy. In the VIPER algorithm decision trees are chosen because their structured nature allows to formally prove properties of the policy they represent when the environments dynamics are known and expressible in closed form.

The main contribution of the paper is the adaptation of VIPER to ensembles of trees, specifically to mixtures of experts, and an adaption of the mixture of experts algorithm to non-differentiable experts. The mixture of experts is extended to decision trees with an em-like two step procedures that alternatively trains the experts and their input dependent weighting function. The main reason for extending VIPER to multiple trees is to increase the representation power of the procedure and obtain more faithful approximations of the original policy. The choice of using a input-dependent linear mixture of trees model instead of a more classical aggregation procedure like random-forests or xg-boosts is instead due to the assumed interpretability of input-dependent linear combinations.

The results in the paper seems to demonstrate comparable or improved performance of MOET over viper, but also show the systematic inability of both methods in imitating the actual neural network policy in all the environments but CARTPOLE. The authors conclude showing that the formal methods proposed to verify the policy in (Bastani et al, 2018) can be extended to MOET as they only rely on the choice of using decision trees as a target model.

On overall the paper creates a consistent narrative grounded on the notions that decision trees and linear models are interpretable, neural networks are not, individual decision trees can only create limited decision boundaries. The authors frame their technical contribution (em-like training of decision trees mixtures of experts) as necessary for creating more sophisticate decision boundaries through ensembling, while keeping the interpretability of the model (the ensembling method is linear and the components are decision trees).

While the narrative is consistent and consequent technical work seems valid the paper has some flaws that put it in my opinion slightly below the acceptance threshold:

1) The notions that linear models and decision trees have the "interpretable" propertiy is simply assumed as a known fact in the scientific field, while it is actually a largely discussed point of contention. See as an example "Lipton 2017, The Mythos of Model Interpretability". The authors should explain specifically what they mean by interpretability and how the properties of decision trees and linear mixtures of experts are consistent with their definition.

2) On the overall paper (similarly to the narrative used for Viper) the authors speak of interpretability as if it was referred to the original deep neural network model while they only interpret the surrogate model. The lack of a formal definition of interpretable masks the problem of interpreting a model through the interpretation of a surrogate model, which would rely on some proof that links whatever is demonstrated on the surrogate to hold for the original model.

3) The presentation of the results focus on the improvement of MOET with respect to VIPER. As a results it is difficult to compare the performance of both procedures to the original neural network model. For Pong and Acrobot the surrogate models do not achieve comparable performance, nor seems to be good imitations of the original policy, the reasons and the implication of this fact are not discussed.

4) There should be a better connection with the knowledge distillation/model compression literature and in general the important idea that the same computation can be represented by different algorithms. (Born Again Trees, Model Compression, Distilling the Knowledge in a Neural Network, Policy Distillation, Born Again Neural Networks).  There is also some related novel literature mapping neural network policies and word models to minimal hidden-markov models that could be of interest to the authors (Learning Finite State Representations of Recurrent Policy Networks, Learning Causal State Representations of Partially Observable Environments).


**Experience Assessment:**

I have published one or two papers in this area.

**Review Assessment: Checking Correctness Of Derivations And Theory:**

I assessed the sensibility of the derivations and theory.

**Review Assessment: Checking Correctness Of Experiments:**

I carefully checked the experiments.

**Review Assessment: Thoroughness In Paper Reading:**

I read the paper thoroughly.

---

> ### Author Response · Authors · 2019-11-09
> **Response to Official Blind Review #2**
>
> Thanks for your review and helpful feedback. Below we provide answers and clarifications related to the different points made in the review.
>
> Answer to Point 1:
>
> We consider interpretability of MOET (and DT) models relative to the deep neural networks. Indeed, Lipton argues that linear models cannot be a priori considered more interpretable than neural networks since they often use heavily engineered features to approach neural networks in performance thus sacrificing decomposability. However, we do not use heavily engineered features and we use simple models. In Lipton's terms, by interpretability we mean transparency, i.e.: 1) simultability, 2) decomposability and partially 3) algorithmic transparency. As noted by Lipton, simultability of decision tree and linear models is preserved because our models are small (2 <= depth <= 10) and we do not use high dimensional features. Similarly, decomposability is preserved because we do not use heavily engineered features. Finally, algorithmic transparency is achieved because MOET relies on DT training for the experts and linear model training for the gate, both of which are well understood. However, transparency is partially achieved because the alternating refinement of initial feature space partitioning and experts makes the procedure more complicated.
>
> More importantly, as the paper describes we have a well-founded translation of MOET models to SMT (Satisfiability Modulo Theories) formulas, which opens a new range of possibilities for interpreting and validating the model using automated reasoning tools. SMT formulas provide a rich means of logical reasoning, where a user can ask the solver questions such as: “On which inputs do the two models differ?”, or “What is the closest input to the given input on which model makes a different prediction?”, or “Are the two models equivalent?”, or “Are the two models equivalent in respect to the output class C?”. Answers to these questions can help better understand and compare models in a rigorous way. Note also our symbolic reasoning of the gating function and decision trees allows us to construct SMT formulas that are readily handled by off-the-shelf tools, whereas direct SMT encodings of neural networks do not scale for any reasonably sized network because of the need for non-linear arithmetic reasoning.
>
> We will add and emphasize both of these points in the revision.
>
> Answer to Point 2:
>
> There seems to be a misunderstanding here probably caused by our presentation. We do not try/claim to interpret the original neural network model, instead we only use it as a proxy to train an interpretable and verifiable MOET model. This is similar in spirit to Viper, which trains a DT policy not as a means of interpreting the DRL policy via the surrogate DT model, but with a goal of constructing a verifiable policy that can be used directly and achieves acceptable performance. We show that with MOET models we can achieve better performance than Viper while still maintaining verifiability and interpretability of the trained MOET models.
>
> Answer to Point 3:
>
> Our goal is not to outperform the original policy, rather our goal is to create a verifiable surrogate with acceptable performance. Note that for a fair comparison between MOET and Viper the paper only shows results that are averaged across multiple trained models. In fact, the best performing MOET models achieve the perfect reward for Cartpole and Pong, solve Mountaincar environment, and achieve close to DRL performance on Acrobot. The following table shows the best case performance of MOET models and original agent (averaged across multiple trials as noted in the paper):
>
>                                  DRL         MOET      MOET_h
>   Cartpole                 200             200              200
>   Pong                         21               21                 21
>   Acrobot              -68.60        -73.64          -73.31
>   Mountaincar   -105.27        -97.57          -97.59
>
> Answer to Point 4:
>
> Thank you for the pointers to the papers. We will add a discussion on these two areas of knowledge distillation and constructing finite state representations of models to our related work section. In summary, the model compression and distillation literature tries to construct student networks that transfer policy or learnt function from a teacher network. But unlike our MOET models, the student networks are still neural networks and therefore less interpretable and more importantly not verifiable. In comparison with learning finite state moore machines, MOET enables learning more hierarchical behaviors because of tree representation and learning complex predicates in a more scalable way.
>
> We hope our clarifications addresses your concerns, but please let us know if there might be any additional questions or clarifications.

---

> > ### Author Response · Authors · 2019-11-13
> > **Feedback to Response**
> >
> > Dear reviewer,
> >
> > Please let us know if our response helped address your concerns regarding the interpretability definition in context of Lipton 2017, use of surrogate model as the final verified policy, comparison with original agent performance, and relationship with distillation and model compression literature. Also, please let us know if there might be any additional questions or clarifications needed for possible reconsideration.
> >
> > Thank you,
> > Authors

---

### Author Response · Authors · 2019-11-12
**Feedback Incorporated In New Paper Version**

Thanks to all the reviewers for their helpful and constructive feedback. We have uploaded a new paper revision to address the comments and feedback:

1. Added a discussion of Knowledge Distillation and Model Compression work to the related work (Section 2).
2. Added a clarification of what is meant by interpretability in the field of machine learning in accordance to the Lipton's: "The Mythos of Model Interpretability" work (Section 2), as well as discussion of how MOET achieves interpretability in those terms (at the end of Section 5).
3. Added tables to Appendix E: Table 2 showing results averaged across multiple MOET configurations, and Table 3 showing results of best performing MOET models.
4. Fixed minor typos and grammar mistakes.
5. Changed description of Table 1 and Figure 2 to be more detailed and thus self-contained.

Please let us know in a case of any additional questions or further suggestions how the paper can be improved.

---

### Decision · Program_Chairs · 2019-12-19

**Decision:**

Reject

**Comment:**

This paper aims at making a deep RL policy interpretable and verifiable by distilling the policy represented by a deep neural network into an ensemble of decision trees. This should be done without hurting the performance of the policy. The authors achieve this by extending the existing Viper algorithm. The resulting approach can imitate the deep policy better compared with Viper while preserving verifiability. Experiments show that the proposed method improves in terms of cumulative reward and error rate over Viper in four benchmark tasks.

The amount of improvement over the original Viper is not convincing given the presented results. Moreover, reviewers uniformly agree that the contribution of this work is incremental. I therefore recommend to reject this paper.